# Targeting Gut Microbiota to Treat Hypertension: A Systematic Review

**DOI:** 10.3390/ijerph18031248

**Published:** 2021-01-30

**Authors:** Joonatan Palmu, Leo Lahti, Teemu Niiranen

**Affiliations:** 1Department of Medicine, University of Turku, FI-20014 Turku, Finland; tejuni@utu.fi; 2Division of Medicine, Turku University Hospital, FI-20521 Turku, Finland; 3Department of Public Health Solutions, Finnish Institute for Health and Welfare, FI-00271 Helsinki, Finland; 4Department of Computing, University of Turku, FI-20014 Turku, Finland; leo.lahti@utu.fi

**Keywords:** blood pressure, dietary sodium, gut microbiota, hypertension, lactobacillus, salt intake

## Abstract

While hypertension remains the leading modifiable risk factor for cardiovascular morbidity and mortality, the pathogenesis of essential hypertension remains only partially understood. Recently, microbial dysbiosis has been associated with multiple chronic diseases closely related to hypertension. In addition, multiple small-scale animal and human studies have provided promising results for the association between gut microbial dysbiosis and hypertension. Animal models and a small human pilot study, have demonstrated that high salt intake, a risk factor for both hypertension and cardiovascular disease, depletes certain *Lactobacillus* species while oral treatment of *Lactobacilli* prevented salt-sensitive hypertension. To date, four large cohort studies have reported modest associations between gut microbiota features and hypertension. In this systematic literature review, we examine the previously reported links between the gut microbiota and hypertension and what is known about the functional mechanisms behind this association.

## 1. Introduction

High blood pressure (hypertension) remains the leading modifiable risk factor for cardiovascular morbidity and mortality [1]. As the prevalence of treatment resistant hypertension is approximately 15%, a need for novel therapeutic modalities for high blood pressure exists [2]. In addition to a shortage of treatment options, the pathogenesis of essential hypertension remains elusive [1,3]. In fact, while increased risk of hypertension has been associated with both adverse lifestyles and over 900 genetic loci, common genetic risk variants explain only <6% of the variance in systolic blood pressure (BP) [4].

Gut microbiota dysbiosis may represent a potentially modifiable risk factor for high blood pressure as it has recently been associated with multiple conditions (e.g., obesity, metabolic syndrome, diabetes, and cardiovascular disease) that are closely associated with hypertension [5]. However, the literature on the potential links between the gut microbiota and blood pressure has not been reviewed to date.

In this systematic literature review, we examine the relation of gut microbiota and hypertension. The specific review questions addressed are: (1) is gut microbiota associated with hypertension, (2) is hypertension associated with functional changes in the gut microbiota, (3) what is the effect size of such interactions, and (4) could gut microbiota modulation be used as a treatment modality for hypertension?

## 2. Materials & Methods

### 2.1. Search Strategy

We performed a systematic literature review for original research articles using Medical Literature Analysis and Retrieval System Online (MEDLINE), Excerpta Medica database (EMBASE), and Cochrane Library. The literature search consisted the intersection of hypertension research (“blood pressure” or “hypertension” in title, author keywords, or Medical Subject Headings [MeSH] keywords) and microbiota studies (“Gastrointestinal Microbiome” MeSH keyword or words microbiota, microbiota or metagenomics in title or abstract). While the focus of the literature review was limited to gut microbiota, the latter search term was given a general form to reduce the false exclusion of relevant research. The literature search was performed on 4 September 2020 without language and publication date restrictions.

### 2.2. Study Selection, Data Extraction, and Data Analysis

Our systematic literature review followed the Preferred Reporting Items for Systematic reviews and Meta-Analyses (PRISMA) guidelines [6]. The inclusion criteria were published original research articles that reported results between sequenced gut microbiota and essential hypertension. In particular, studies focusing on associations for diseases other than hypertension (e.g., renal disease, liver disease, or sleep apnea) were excluded. Studies on oral microbiota were also excluded, due to the expected differences in the environmental conditions between the oral cavity and gut. We also deemed the studies between the maternal microbiota and offspring hypertension to be outside the scope of our literature review. Finally, we included the articles with the following study interventions: stool transfer, orally administrated probiotics, sodium, antihypertensive medication, and short chain fatty acids (SCFAs), or genomic knockout models.

Duplicate search results were automatically detected using Digital Object Identifiers. Manual screening was performed by single author using first titles, second abstracts, and finally full-text. Included manuscripts were mutually non-exclusively grouped in animal models and human studies. In animal models, the effect of interventions to circulating metabolites, vasculature, gut wall, and gut content are presented in a summary table. In human studies, the associations observed between hypertension, high dietary salt, gut microbiota, and SCFAs are presented in a second summary table. When original research articles reported associations between a large number of different groups or between multiple interventions, the one most suited to answer to previously described attributes was selected. A preliminary literature review did not reveal compatible numerical properties in published gut microbiota-hypertension studies that could be combined to perform a meta-analysis. We used in the review two software tools implemented in R. Revtools [7] was used to conduct the article screening and PRISMAstatement [8] to draw the flow charts of study inclusions and exclusions.

## 3. Results

The literature search identified 669 potential original research articles. We excluded 180 duplicate search results, 264 manuscripts after title screening, and 132 manuscripts in abstract screening. In full-text screening, we excluded four incomplete clinical trials, four review articles, 20 manuscripts missing gut microbiome sequencing and 27 manuscripts due to the intervention type. Finally, we included 38 original research articles in our literature review (Figure 1). We summarize in the following section the 22 animal studies that examined the potential causal relation between gut dysbiosis and hypertension (Table 1). These promising results from animal models have given rise to 13 small studies, four large cross-sectional epidemiological studies and one interventional pilot study on the association between gut dysbiosis and human hypertension (Table 2).

### 3.1. Microbial Data Is Compositional in Nature

The microbial abundances share a subtle but highly consequential property imposed by limits of high-throughput sequencing. The observed count of one species affects the observed counts of all other species making the data inherently compositional (Figure 2) [43]. The experimental setup of high-throughput sequencing resembles the frequent example used in the teaching of probability calculation where a fixed number of different colored marbles are removed from a bag of unknown number of marbles. When the microbial load (total number of marbles) is not known, the data provides only information about the relative proportions of studied characteristics and using inappropriate statistical methods can lead up to a 100% false discovery rate [44]. One popular method to render microbial counts comparable across samples is the centered log-ratio transformation where bacterial counts are divided by the geometric mean of all observed counts. [43] In addition to appropriate statistical methodology, flow cytometry could be used to estimate the cell counts (while not total microbial load) improving the interpretation of 16S rRNA sequenced data in particular [44,45].

### 3.2. Animal Studies

#### 3.2.1. Changes in Gut Microbiota Are Associated with Hypertension in Rodents

Fecal microbiota transplantation from hypertensive donors to normotensive rats has been demonstrated to increase BP and induce changes in the abundances of multiple gut microbial species [9,19,24,25]. Notably, cross-species fecal microbiota transplantation from hypertensive humans to mice promotes hypertension [17]. While various associations between BP and gut microbiota have been observed (Table 1), *Lactobacillus* abundances appear to be particularly susceptible to sodium in salt-induced hypertension [27,31]. The association between probiotic intake and blood pressure has been studied in experimental animal models. In mice, *L. fermentum* treatment prevents hypertension and reduces endothelial dysfunction [23]. Consistently, long-term administration of *Lactobacillus fermentum* or *L. coryniformis* plus *L. gasseri* have been demonstrated to reduce systolic BP and decrease vascular inflammation in spontaneously hypertensive rats [15]. Increased nitric oxide production may play a critical role in the beneficial (antihypertensive) effect of *Lactobacilli* [20].

BP has a well-defined morning diurnal rhythm peak, dropping 10–20% at rest [11]. There is some evidence that gut microbiota shares synchronous time-of-day variation with BP that also correlates with the levels of circulating renal markers [11]. In rats, microbial pathways characterized with biosynthesis were upregulated at active phase of day (nighttime) while metabolite degradation pathways were upregulated at rest (daytime) [11]. This finding may have an impact on clinical hypertension care also as reduced nocturnal systolic BP dipping has been associated with an increased risk of cardiovascular events and diurnal variation of BP may affect both medication timing and microbial-targeted strategies [11,46].

#### 3.2.2. Increased BP Is Associated with Inflammation and Gut Wall Pathology

The onset of hypertension has been associated with adrenergic nervous system activation that becomes more pronounced with increasing BP [47]. Dietary habits and psychosocial environment (stress) are possible drivers of this sympathetic activity that could alter host-microbiota cross-talk perpetuating peripheral and neural inflammation [22]. In hypertensive rats, green fluorescent protein staining demonstrated robust retrograde labeling from small intestine to the paraventricular nucleus of the hypothalamus compared to normotensive rats [2]. Changes in gut microbiota also increase plasma noradrenaline concentration (a marker of sympathetic activity), altered T_H_17/Treg balance in mesentric lymph nodes, and levels of pro-inflammatory cytokines (tumor necrosis factor-α [TNF-α], interleukin [IL]-1β, IL-6, IL-17a, and interferon-γ) in brain paraventricular nucleus [24,25]. In addition, increased tyrosine hydroxylase (a rate-limiting enzyme in the synthesis of norepinephrine) immunoreactivity was observed in the small intestine of hypertensive compared to normotensive rats [2]. In a model of angiotensin II-induced hypertension, infusion of an anti-inflammatory agent in the cerebral ventricles alleviated the activation of microglia (resident macrophages), decreased the concentration of proinflammatory cytokines (IL-1β, IL-6, TNF-α) in the paraventricular nucleus, and induced changes in several gut microbial communities [22].

In addition to an augmented paraventricular nucleus-gut connection and increased markers of inflammation, hypertension has been associated with increased intestinal permeability and reduced mRNA levels of intestinal gap junction proteins in rats (Table 1) [2,24]. In addition, spontaneously hypertensive rats (SHRs) have lower mucin transcripts levels, the gel-like protective barrier proteins, compared to Wistar Kyoto rats [21]. Circulating bacterial wall components, such as lipopolysaccharides, can activate vascular toll-like receptors contributing to low-level chronic inflammation exacerbating hypertension [21]. However, treatment with renin-angiotensin system inhibitors has been reported to ameliorate gut wall pathology [28,30]. Candesartan treatment increases the intestinal expression of genes encoding tight junction proteins and serum levels of lipopolysaccharides-binding protein in SHRs [28]. Captopril treatment is associated with an increased number of goblet cells, increased villi length, reduced ileal fibrosis and reduced gut wall permeability serum markers in SHRs [30]. Intracerebroventricular administration of an anti-inflammatory agent also prevents gut wall pathology by reducing fibrosis and thickness of muscularis layer in the gut, and increasing villi length and the number of goblet cells [22]. In summary, hypertension has been associated with gut wall pathology and augmented paraventricular nucleus -gut connectivity in animal models.

#### 3.2.3. Changes in Gut Microbiota Are Associated with Circulating Metabolites in Animal Models

SCFAs are microbial fermentation products of undigested carbohydrates that have the ability to enter host circulation (Table 1) [48]. In germ-free animals, a 100-fold reduction in cecal and circulating SCFA levels has been reported compared to conventional animals [48]. In particular, three major SCFAs, acetate, propionate, and butyrate, have been reported to have borderline undetectable concentrations in the plasma of the germ-free mice [49].

Hypertension has been associated with a decreased numbers of acetate- and butyrate-producing bacteria [20]. Angiotensin II type 1 receptor blocker treatment of SHRs has been reported to increase fecal acetate, propionate, and butyrate levels [28]. In an animal model of mineralocorticoid excess, dietary acetate increased the relative abundance of acetate-producing gut bacteria while reducing blood pressure and cardiac hypertrophy compared to untreated mice [18]. Acetate and butyrate treatment has been reported to alleviate the gut-wall pathology and lower BP in hypertensive rats and mice while no changes were observed in normotensive rats [16,21].

The SCFAs have been demonstrated to affect renin release in juxtaglomerular cells and modulate BP through G-protein coupled receptors [48]. A family of SCFA binding G-protein coupled receptors has been recently discovered [26]. A model of genomic excision of G-protein coupled estrogen receptor has given support to host-microbial cross-talk demonstrating that host genomic change modulates gut microbiota which in turn affects host BP [26]. The role of gut microbiota on the circulating metabolites was further studied in model of germ-free and conventional mice [12]. While hypertension induced changes were observed in the fecal and plasma metabolites of conventional mice, no changes were observed in germ-free mice [12]. In conclusion, the host and gut appear to share two-way interaction which is in part governed by SCFAs.

#### 3.2.4. High Dietary Salt and Gut-Immune Axis in Animal Models

The deleterious effect of high dietary salt on cardiovascular health has been recently associated with the gut-immune axis [27]. High salt intake has been reported to modulate gut microbiota, particularly depleting *L. murinus* in mice (4% dietary and 1% drinking water NaCl vs. 0.5% dietary NaCl for 14 days) while the growth of various *Lactobacilli* was inhibited by sodium (half maximal growth inhibition 0.6 mol/L in vitro vs. 0.3 mol/L colonic NaCl concentration under high dietary salt) [27]. *L. murinus* administration reduces consistently salt-sensitive hypertension and prevents the increase in IL-17A producing CD4^+^ T_H_17 among small intestinal, colonic and splenic lamina propria lymphocytes [27]. The changes in gut microbiota in hypertension have been associated with increased T_H_17/Treg ratio in mesenteric lymph nodes and activation of adaptive immune response [25]. The B7 ligand co-stimulated T cell activation and modulation of T_H_17/IL17 axis have been proposed to share an essential role in the development of endothelial dysfunction, increased vascular oxidative stress, and hypertension upon fecal microbiota transplantation from SHR to Wistar Kyoto rats [25].

Salt-induced hypertension has been demonstrated to increase the number of immune cells in mesenteric arterial arcade and aorta (CD45^+^, CD3^+^, CD4^+^, and CD8^+^ T cells) [14]. In addition to the decrease of *Lactobacillus* species, various gut microbial changes have been associated with high dietary salt (4–8% dietary NaCl; Table 1) [10,13,14,31]. A novel mechanism for salt-induced hypertension has been proposed: salt induced changes in gut microbiota upregulate the production of corticosterone which enters circulation causing mineralocorticoid excess (pseudoaldosteronism), leading to hypertension, hypokalemia and inhibition of aldosterone synthesis [29].

### 3.3. Human Studies

#### 3.3.1. Gut Microbiota Is Associated with Human Hypertension

Numerous scientific publications have reported changes in gut microbial abundances and serum metabolites between groups of normotensive, pre-hypertensive and hypertensive individuals (Table 2) [17,35,37,38,39,42]. In particular, Black hypertensive individuals have been reported to have higher BP, greater prevalence of treatment resistant hypertension, increased pro-inflammatory potential in gut microbiota, and greater oxidative stress markers in plasma as compared to White hypertensive individuals [41]. Even subtypes of hypertension, namely isolated systolic and diastolic hypertension, share distinct gut microbiota profiles [33].

Four large cross-sectional studies have reported associations between gut microbiota and human hypertension. In the TwinsUK (N = 2737, 89% female, age 60 ± 12; 16S rRNA; record identified through other sources) study, no associations were observed between 68 various microbiota markers and self-declared hypertension after correcting for multiple testing [50]. In the Coronary Artery Risk Development in Young Adults (CARDIA; N = 529, 54% female, age 55 ± 3; 16S rRNA) study, a negative association was observed both for microbial alpha diversity and the abundance of *Robinsoniella*-genus with systolic BP [40]. In the FINRISK 2002 study (N = 6953, 55% female, age 49 ± 13; shotgun metagenomics), 45 microbial genera and 19 *Lactobacillus* species were associated with BP indices [5]. In the HEalthy Life In an Urban Setting (HELIUS; N = 4672, 52% female, age 50 ± 12; 16S rRNA) study, gut microbiota explained 4.4% of the overall unadjusted systolic BP variance but the proportion of variance explained was strongly divergent in different ethnic groups (4.8% for Dutch and <0.8% for others) [1]. In previous studies, the observed association between gut microbiota and hypertension was insignificant for British [50], African [1], Ghanaian [1], South Asian [1], and Turkish participants [1]. While significant associations have been observed in American [40], Finnish [5], Moroccan [1], and Dutch [1] participants, only the gut microbiota of the Dutch cohort demonstrated a high level of explained variance with regard to BP [1]. Altogether, it appears that associations between gut microbiota and BP exist, but they vary between different ethnic populations.

#### 3.3.2. High Dietary Salt Is Associated with Gut Microbiota in Human Hypertension

While the deleterious effect of high-salt diet to BP and cardiovascular health is well documented, most studies have focused on the pathophysiology of the high-salt diet in kidneys, vasculature, and the sympathetic nervous system [27]. To study the effect of high-salt diet on gut microbiota, a pilot study (N = 12) was performed in healthy males receiving slow-releasing 6 g sodium chloride supplementation (total salt intake 13.8 ± 2.6 g/day for 14 days) [27]. Sodium supplementation resulted in an increase in nocturnal BP, increase in peripheral blood CD4^+^ IL-17A^+^TNF-α^+^T_H_17 cells, and reduction of *Lactobacillus* species (Figure 3) [27]. In the FINRISK 2002 study (N = 6953; cross-sectional study), genus-level *Lactobacillus* was not associated with BP while positive and negative associations of multiple *Lactobacillus spp.* and BP were detected [5]. In particular, *L. paracasei* was negatively associated with BP, whereas *L. salivarius* was positively associated with only pulse pressure [5]. In addition, *L. paracasei* was negatively, and *L. salivarius* was postitively associated with dietary sodium intake (N = 829 for 24-h urinary sodium subsample) [5]. *Lactobacilli* are not a dominant member of the human feces, observed in only 15–42% of individuals [5,27]. *Prevotella* and *Bacteroides* have been positively associated with high-salt diet in human [14]. In the FINRISK 2002 study, *Bacteroides* was positively associated with BP and *Prevotella* had a borderline significant negative association with diastolic BP [5]. In summary, while gut bacteria demonstrate consistent links with high-salt diet and hypertension, genus level abundances may offer too generalized and even contradictory information to capture all the relevant associations.

#### 3.3.3. SCFAs Are Altered in Human Hypertension

Hypertensive individuals are reported to have decreased levels of circulating SCFAs, such as acetate, isobutyrate, buryrate and isovalerate, and increased levels of fecal SCFAs, such as acetate, butyrate, propionate, and valerate, compared to normotensive individuals [1,16,32,34,36]. The relationship between the number of SCFA-producing gut bacteria and host SCFA levels appears to be convoluted [32]. In particular, BP was negatively associated with fecal SCFA-producing bacteria, but positively associated with fecal SCFA levels in the HELIUS study [1]. While trimethylamine N-oxide, a product of bacterial metabolism derived from phosphatidylcholine (found in red meat, dairy products, eggs, and fish) has been associated with atherosclerosis and cardiovascular health, it was not related to hypertension in a small human study [32]. All in all, the previous studies demonstrate changes in SCFA production and absorption in human hypertension [1,32].

#### 3.3.4. Gut Microbiota May Modulate Host Inflammatory Response

Similar to animal models, human hypertension is associated with increased intestinal wall permeability [16]. Mucosal dendritic cells detect local pathogens and regulate intestinal immune homeostasis [14]. High-salt diet has been associated, in addition to changes in circulating SCFAs, with increased formation of isolevuglandins in CD11c^+^ antigen-presenting cells leading to T cell activation and production of IFN-γ [14]. While the effect of SCFAs is governed by the activated receptor, SCFAs are linked to important anti-inflammatory effects suppressing the production of TNF-α and IL-6 [3]. In particular, IL-17A producing CD4^+^ T_H_17 cells are reported to transmit immune reaction in hypertension [16,27]. The gut microbiota may therefore modulate host inflammatory response and BP also in humans.

## 4. Conclusions

### 4.1. Summary

Several animal studies have suggested that gut microbiota dysbiosis and hypertension could be causally related. These results are in part supported by results from four large-scale observational epidemiological studies and from one interventional pilot study performed in humans.

Although several studies have reported on the potential association between the Firmicutes/Bacteroidetes ratio and blood pressure, [9,11,13,18,21,23,28,39] this approach has several challenges. These challenges include the lack of information on which of the two taxa is the main driver of the ratio and the use of relatively heterogenic and coarse phylum-level information. We have therefore refrained from focusing on these studies in this review.

Although variation in the gut microbiota has been associated with BP, the proportion of BP variance explained by gut microbiota seems to be only modest in most human cohort studies (<1%), when accounting for potential confounding factors. In addition, the observed associations vary greatly in different ethnic populations. However, the key gut microbial characteristics (diversity indices, microbial abundances, and proportions of variance) vary by study either due to technical factors or due to biological variation, which makes comparing and replicating results across studies challenging. Gut microbiota has been proposed to demonstrate reproducible patterns of variation, coined enterotypes, that could offer an useful tool to simplify the complexity of the gut ecosystem and possibly simplify studying the comorbidity of hypertension [51]. Due to the large number of bacterial species in gastrointestinal tract and limitations of available taxonomic resolution, most studies have only studied genus level abundances which may offer overly general information that only partially captures the interactions between the gut and the host. Our recent preprint demonstrates that increased taxonomic resolution improved the predictive performance of machine learning models in liver disease and proposes the use of species or even greater levels of resolution to improve our understanding of the role of gut microbiota to human health [52].

### 4.2. Future Directions

To assess the role of SCFAs in host, careful consideration is necessary to differentiate the associations reported for SCFA producing bacteria, fecal SCFA levels, and circulating (bioactive) SCFA levels. Finding methods for accurate, absolute scale and possible non-direct measurement of the gut microbiota and its functional pathways could offer improvements in studying the microbiota-BP associations in large-scale epidemiological studies. In addition, human studies could be improved: (1) by using more accurate methods of BP measurement, such as home or ambulatory BP; (2) by performing deep metagenomic sequencing that would study species and strain level abundances; and (3) by implementing large-scale interventional studies that aim to manipulate dietary sodium intake and gut *Lactobacillus* species to establish a causal relationship between the gut microbiota and BP in humans.

## Figures and Tables

**Figure 1 ijerph-18-01248-f001:**
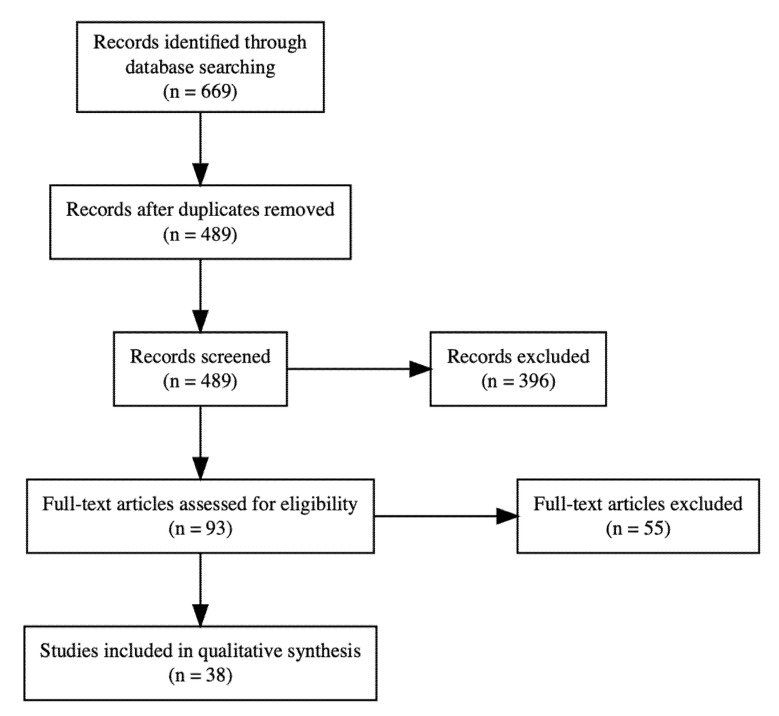
PRISMA flow chart.

**Figure 2 ijerph-18-01248-f002:**
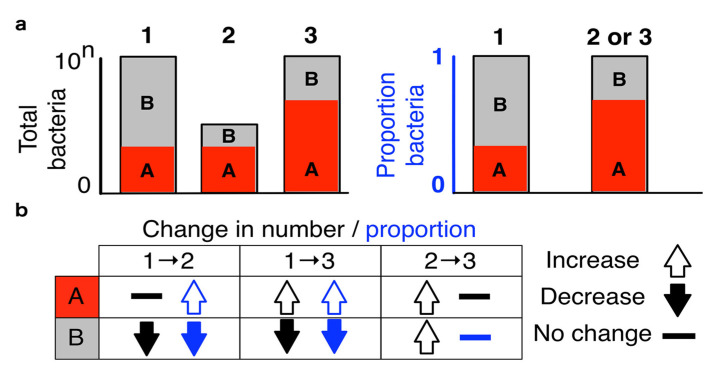
Gut metagenome sequencing data are compositional. In this illustration, we observe three samples (1, 2 and, 3) and two microbial species (A and B). The sequence counts of microbial samples do not provide information about the absolute microbial abundances (black) making the observed data inherently compositional (blue). Therefore, real change in the total abundance of one microbial species can impose perceived change in other observed species (bottom left). Conversely, multiple proportionate changes in the real microbial abundances may be hidden from the observed abundances (bottom right). Adapted from Gloor et al. [43]. Licensed under Creative Commons Attribution License (CC BY, 2017).

**Figure 3 ijerph-18-01248-f003:**
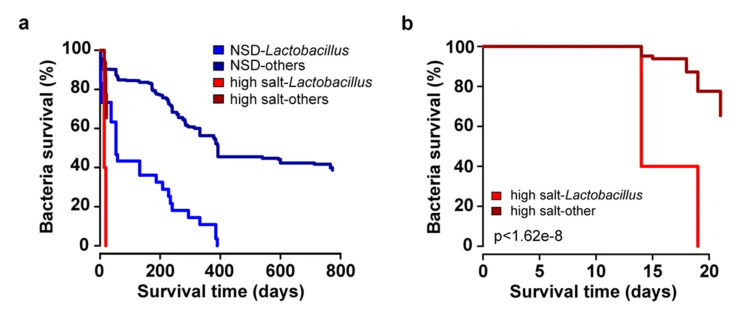
Effects of high-salt challenge in healthy human subjects. The left panel displays Kaplan-Meier survival curves for gut bacteria in individuals under high-salt intervention (red; N = 12) and healthy controls (blue; N = 121). The bright shades of red and blue indicate the *Lactobacillus* species while the dark shades indicate the set of all other detected gut bacteria. The right panel displays the results for the salt intervention group during the first 20 days of the intervention. In summary, high-salt challenge decreases the abundances of several gut microbial species while the loss of *Lactobacillus* species is particularly rapid. NSD, normal salt diet. Adapted by permission from the Springer Nature: Nature, Salt-responsive gut commensal modulates TH17 axis and disease, Wilck et al. [27].

**Table 1 ijerph-18-01248-t001:** The effect of rodent interventions to circulating metabolites, vasculature, gut wall, and gut content.

Study	Animals	Sequencing	Intervention	Increased or Uppregulated in Serum, Plasma, Vasculature, or Organs	Decreased or Downregulated in Serum, Plasma, Vasculature, or Organs	Increased or Uppregulated in Gut	Decresased or Downregulated in Gut
Adnan [9]	WKY rats	16S rRNA (V4)	FMT from SHRSP			Erysipelotrichaceae, Dorea, Anaerostipes, Bacteroidales, Micrococcaceae, Ruminococcus, Deferribacterales, Deferribacteres, Mucispirillum, Deferribacteraceae, Deferribacteres, Lactococcus, Desulfovibrio, Deltaproteobacteria, Desulfovibrionales, Roseburia, Coprococcus, Lachnospiraceae, Clostridiales, Firmicutes	Bacteroidetes, Bacteroidia, Erysipelotrichi, Erysipelotrichales, Allobaculum, Actinobacteria, Bacteroidaceae, Bacteroides, Actinobacteria, Bifidobacteriales, Bifidobacterium, Enterobacteriales, Gammaproteobacteria, Enterobacteriaceae, Betaproteobacteria, Sutterella, Alcaligenaceae, Bacillales, Bacillaceae, Coprobacillus, Coriobacteriales, Coriobacteriia, Adlercreutzia, Holdemania, Enterococcus
Bier [10]	DSS rats	16S rRNA (V4)	4% NaCL			*Erwinia*, *Christensenellaceae*, *Corynebacteriaceae*, acetate, propionate, isobutyrate	*Anaerostipes*
Chakraborty [11]	DSS rats	16S rRNA (V3-V4)	Diurnal timing dependent alterations or disrupted circadian rhythm			Increases in rest: Streptococcaceae, Veillonellaceae, Clostridiaceae (negative association with SBP), Helicobacteriaceae, Lactobacillus	Decreases in rest: Sutterella (positive association with SBP), Erysipelotrichaceae
Cheema [12]	C57BL/6 and GF mice	16S rRNA (V3-V4)	Angiotensin II	4-ethylphenyl sulfate (GI), p-cresol sulfate (GI), p-cresol glucuronidarke (GI), spermidine	Mannose, trans-4-hydroxyproline, xylose, N,N,N-trimethyl-5-aminovalerate (GO), indoleacetate, ferulic acid 4-sulfate, gentisate (GO), indoleacetylglycine (GO)	*Anaeroplasmataceae*; 25 uppregulated metabolites	*Lachnospiraceae*; 71 downregulated metabolites (including N,N,N-trimethyl-5-aminovalerate, spermidine, indoleacetate, trans-4-hydroxyproline, xylose)
Chen [13]	Sprague-Dawley rats	16S rRNA (V3-V4)	4% NaCl (20% fructose in water)	Renin, Ang-II		*Rikenellaceae*	*Desulfovibrionaceae*
Ferguson [14]	C57Bl/6 mice	16S rRNA (V4)	8% NaCl			*Firmicutes*, *Prevotella*, mesenteric arterial immune cells (CD45+, CD3+, CD4+, and CD8+ T cells)	*Bacteroidetes*, *Leuconostoc*, *Streptococcaceae*, *Lachnospiraceae UCG-06*, *Lachnospiraceae FCS020*
Gómez-Guzmán [15]	SHR	qRTi-PCR	K8 and LC9	Vascular relaxation induced by acetylcholine	Heart weight, ventricular weight and kidney weight relative to body weight	Lactobacillus	Bacteriodes, Clostridium
Kim [16]	C57B16 mice	16S rRNA (V3-V4)	Butyrate		Heart rate and increased coronary flow	*Akkermansia muciniphila*; mRNA for occludin, ZO-1, and claudin 4; CCR2^+^CD4^+^Il-17^+^Th17 cells	
Li [17] (2017)	GF C57BL/6L mice	16S rRNA (V4)	FMT from hypertensive human			*Coprobacillus*, *Prevotella*	*Anaerotruncus*, *Coprococcus*, *Ruminococcus*, *Clostridium*, *Roseburia*, *Blautia*, *Bifidobasterium*
Marques [18]	C57Bl/6 mice	16S rRNA (V3-V4)	Acetate		Cardiac fibrosis and left ventricular hypertrophy	*Bacteroides acidifaciens*	
Mell [19]	DSS rats	16S rRNA (V1-V3)	FMT from salt-resistant DSS	Acetate and heptanoate			*Veillonellaceae*
Robles-Vera [20] (2018)	Wistar rats	16S rRNA (V3-V4)	L-NAME	Aortic ring NADPH oxidase activity, TNF-α, IL-1β, and RORγ		*Propionibacterium, propionate*; Th17/Treg activity in mesenteric lymph nodes	Parabacteroides, Bifidobacterium, Olivibacter, Dysgonomonas, Pedobacter, Flavobacterium, and Desulfotomaculum; expression of barrierforming junction transcripts in the colon
Robles-Vera [21] (2020)	SHR	16S rDNA (V4-V5)	Acetate	Vascular relaxation induced by acetylcholine	Left ventricle weight relative to body weight; (microbial) endotoxins	zonula occludens-1, occludin, mucin-2, mucin-3, IL-18, Treg	*Lactobacillus*, _Peptostreptococcaceae, Th17
Santisteban [2]	Sprague Dawley rats (SHR)	16S rDNA (V4-V5)	Angiotensin II	FITC-dextran		Fibrotic area and musuclaris layer in small intestine; CD3+ T-cells, CD68+ macrophages, Iba1+ macrophages; *Streptococcus* (SHR)	levels of tight junctional proteins (occludin, tight junction protein 1, cingulin); *Bifidobacterium* (SHR)
Sharma [22]	SHR (Ang II)	16S rDNA (V4-V5)	Chemically modified tetracycline-3			Ruminococcus and Oscillospira; number of goblet cells and villi length	Proteobacteria, Parabacteroides and Blautia; thickness of the muscularis layer and fibrotic area
Toral [23] (2018)	C57Bl/6J mice	16S rRNA	*LC40*			*Bifidobacterium*, (*Lactobacillus fermentum CECT5716*)	*Anaerostipes*, *Hespellia*, *Prevotella*
Toral [24] (2019a)	WKY rats	16S rDNA (V4-V5)	FMT from SHR	lipopolysaccharides		Bacteroidia, Bacteroidales, Bacteroidetes, Odoribacter, Odoribacteraceae, Coprococcus, Turicbacteraceae, Turicibacterales, Turicibacter; TNF-α, tyrosine hydroxylase, and noradrenaline mRNA	Blautia, Enterococcaceae, Enterococcus; zonula occludens-1 and mucin-2 mRNA
Toral [25] (2019b)	WKY rats	16S rDNA (V4-V5)	FMT from SHR	Aortic NADPH oxidase, TNF-α, INFγ, IL-17, RORγ	Aortic relaxation induced by acetylcholine	CX3CR1, Itga4, and CCR9 in mesentric lymph nodes	
Waghulde [26]	DSS rats	16S rRNA (V1-V3)	Deletion of *Gper1*			Acidovorax, Aeromonadaceae, Bacteroides, Enterococcus, Methylophilaceae, Pseudomonas, Variovorax paradoxus; response to endothelium-dependent vasorelaxation (ACh)	Clostridiaceae, Fusobacterium, Lactobacillus, Pediococcus, Turicibacter
Wilck [27]	FVB/N mice	16S rDNA (V4-V5)	4% NaCL (1% in water)			*Parasutterella; CD4^+^RORγt^+^T_H_17*	*L. murinus*, *Lactobacillus*, *Oscillibacter*, *Pseudoflavonifractor*, *Clostridium XIVa*, *Johnsonella*, *Rothia*; indole-3-lactic acid, indole-3-acetic acid
Wu [28]	SHR	16S rDNA	ATR blocker	LBP		*Lactobacillus*; acetic acid, propionic acid, butyric acid, isobutyric acid, valeric acid, and isovaleric acid; cingulin, occludin and tight junction protein 1	
Yan [29]	Wistar rats	16S rRNA (V3-V4)	8% NaCL (3% in water)	cortisone	aldosterone	corticosterone and enzymes of corticosterone synthesis (Cyp11a1, Cyp11b1, and Hsd11b11)	*B fragilis YCH46*; arachidonic acid in HSD; enzyme of cortisone inactivation (Hsd11b2)
Yang [30]	SHR	16S rRNA (V4-V5)	ACE inhibitor		Marker of gut permeability (I-FABP)	*Firmicutes*, *Proteobacteria*, *Actinobacteria*; number of goblet cells, villi length	*Bacteroidetes*; Fibrotic area
Zhang [31]	C57BL/6J mice	16S rRNA (V4-V5)	4% NaCl	Ang I, glucose, albumin, total proteins, sodium	LDL-C		*Lactobacillus*

ACE, angiotensin-converting enzyme; Ang, angiotensin; ATR, Angiotensin II type 1 receptor; BFM, *Bifidobacterium breve* CECT7263; DSS, Dahl salt-sensitive; GF, Germ free; GO, denotes metabolite product associated with microbial origin; GPR, G protein-coupled receptors; FMT, fecal microbiota transplant; HT, hypertension; K8, *Lactobacillus coryniformis* CECT5711; L-NAME, NG-nitro-L-arginine methyl ester (nitrogen oxide synthase blocker); LBP, lipopolysaccharide-binding protein. LC40, *Lactobacillus fermentum* CECT5716; LC9, *Lactobacillus gasseri*; LDL-C, low-density lipoprotein cholesterol; SHRSP, spontaneously hypertensive rats; SHRSP, spontaneously hypertensive stroke prone rats; T_H_, helper T cell.

**Table 2 ijerph-18-01248-t002:** The associations observed between human hypertension, high dietary salt, gut microbiota, and SCFAs.

Study	Population	Sequencing	Enriched in Hypertension or Positively Associated with BP Indices	Enriched in Normotension or Negatively Associated with BP Indices	SCFA in Hypertension	High Salt Diet
Calderón-Pérez [32]	29 non-treated HT and 32 NT subjects in Reus, Spain	16S rRNA	Bacteroides coprocola, Bacteroides plebeius; Lachnospiraceae	Ruminococcaceae NK4A214, Ruminococcaceae UCG-010, Christensenellaceae R-7, Faecalibacterium prausnitzii, Ruminococcaceae hominis	Decreased serum acetate, isobutyrate, buryrate and isovalerate; increased stool acetate, priopionate, butyrate, valerate	
Dan [33]	67 HT and 62 NT subjects in Beijing, China	16S rRNA (V3-V4)	*Acetobacteroides*, *Alistipes*, *Bacteroides*, *Barnesiella*, *Butyricimonas*, *Christensenella*, *Clostridium sensu stricto*, *Cosenzaea*, *Desulfovibrio*, *Dialister*, *Eisenbergiella*, *Faecalitalea*, *Megasphaera*, *Microvirgula*, *Mitsuokella*, *Parabacteroides*, *Proteiniborus*, *Terrisporobacter*	*Acetobacteroides*, *Acidaminobacter*, *Adlercreutzia*, *Anaerotruncus*, *Asteroleplasma*, *Bulleidia*, *Cellulosilyticum*, *Clostridium III*, *Clostridium IV*, *Clostridium XlVa*, *Coprobacter*, *Enterococcus*, *Enterorhabdus*, *Flavonifractor*, *Gemmiger*, *Guggenheimella*, *Intestinimonas*, *Lachnospiracea_incertae_sedis*, *Lactivibrio*, *Lactobacillus*, *Macellibacteroides*, *Marvinbryantia*, *Olsenella*, *Paraprevotella*, *Parasutterella*, *Phascolarctobacterium*, *Prevotella*, *Romboutsia*, *Ruminococcus*, *Sporobacter*, *Sporobacterium*, *Sutterella*, *Vampirovibrio*, *Veillonella*, *Victivallis*		
de la Cuesta-Zuluaga [34]	441 subjects in Colombia	16S rRNA (V4)			Increased fecal acetate, propionate, and butyrate	
Ferguson [14]	39 subjects with normal and 93 with high sodium intake in USA	16S rRNA	*Prevotella*			Increase in Firmicutes, Proteobacteria, Prevotella, Ruminococcaceae, and Bacteroides
Han [35]	99 non-treated HT, 56 pre-HT and 41 NT subjects in Kailuan, China	Shotgun metagenomic	Streptococcus virus phiAbc2, Salmonella phage vB SemP Emek, Mycobacterium phage Toto	Cronobacter phage CR3, Cnaphalocrocis medinalis granulovirus		
Huart [36]	38 HT, 7 borderline HT, and 9 NT male subjects in Belgium	16S rDNA (V1-V3)			Increased stool acetate, butyrate, and propionate	
Kim [16]	22 HT and 18 NT subjects in Florida, USA	Shotgun metagenomic	Parabacteroides johnsonii, Klebsiella, Anaerotruncus, Eubacterium siraeum, Alistipes indistinctus, Prevotella bivia, Ruminococcus torques, Alistipes finegoldii assciated	Bacteroides thetaiotaomicron, Paraprevotella clara, Paraprevotella	Decrease in plasma butyrate	
Li [17] (2017)	99 HT, 56 pre-HT and 41 NT subjects in Tangshan, China	Shotgun metagenomic	*Acidiphilium*, *Faecalibacterium*	Anaerotruncus, Ruminiclostridium, Robinsoniella, Clostridium, Intestinimonas, Butyricicocous, Pseudoflavonifractor, Paenibacillus, Subdoligranulum, Treponema, Holdemania, Roseburia, Butyrivibrio, Oscillibacter, Marvinbryantia, Akkermansia, Oribacterium, Pyramidobacter		
Li [37] (2019)	104 HT, 63 non-treated HT, 26 NT with hyperlipidemia and 42 NT subjects in Xinxiang, China	16S rRNA (V3-V4)	*Lactococcus*, *Alistipes*, *Subdoligranulum*, *Megasphaera*, *Megamonas*	*Clostridium sensu stricto 1*, *Romboutsia*, *Erysipelotrichaceae UCG.003*, *Ruminococcus 2*, *Intestinibacter*		
Liu [38]	94 HT and 94 NT subjects in Xianyang, China	Bacteria primers	*Eubacteriumm rectale*	*Bifidobacterium*, *Bacteroides thetaiotaomicron*		
Mushtaq [39]	50 HT and 30 NT subjects	16S rRNA (V3/V3-V4), Bacteria primers	*Prevotella_9_*, *Megasphaera*, *Parasutterella*, *Escherichia-Shigella*, *Phascolarctobacterium faecium*	*Faecalibacterium prausnitzii*, *Bacteroides uniformis*		
Palmu [5]	3291 HT and 3662 NT subjects with an urinary sodium subsample of 829 in Finland	Shotgun metagenomic	Acidaminococcus, Actinomyces, Anaerostipes, Bacteroides, Blautia, Cellulomonas, Clostridioides, Collinsella, Coprococcus, Desulfovibrio, Dialister, Dielma, Dorea, Eisenbergiella, Enorma, Enterobacter, Erysipelatoclostridium, Faecalitalea, Holdemania, Intestinibacter, Lactococcus, Megasphaera, Mitsuokella, Paraprevotella, Phascolarctobacterium, Ruthenibacterium, Sanguibacteroides, Sutterella, Turicibacter	Adlercreutzia, Alloprevotella, Anaerotruncus, Coprobacillus, Faecalicoccus, Fournierella, Hungatella, Parasutterella, Prevotella, Sellimonas, Senegalimassilia, Solobacterium, Tyzzerella		Increase in *Lactobacillus salivarius*, decrease in *Lactobacillus paracasei*
Silveira-Nunes [3]	48 HT and 32 NT subjects in Minas Gerais, Brazilia	16S rRNA (V3-V4)	*Lactobacillus salivarius*, *Bacteroides plebeius*, *Eggerthella*	*Roseburia faecis*, *Faecalibacterium prausnitzii*, *Parabacteroides distasonis*, *Fusobacterium*, *Coprobacillus*		
Sun [40]	183 HT and 346 NT subjects in USA	16S rRNA (V3-V4)	*Catabacter*, *Robinsoniella*			
Verhaar [1]	1937 HT and 2735 NT subjects in Amsterdam, Netherlands	16S rRNA (V4)	*Streptococcus*	*Roseburia*, *Clostridium sensu stricto 1*, *Roseburia hominis*, *Romboutsia*, *Ruminococcaceae*, *Enterorhabdus*	Higher stool acetate and propionate	
Walejko [41]	10 black HT, 12 white HT, 10 black NT, and 20 white NT in USA	Shotgun metagenomic				
Wilck [27]	12 healthy males with 6 g sodium chloride intervention in Berlin, Germany	Shotgun metagenomic				Decrease in L. salivarius, L. rhamnosus, L. plantarum,L. delbruenckii, L. casei, L. brevis
Yan [42]	60 HT and 60 NT subjects in China	Shotgun metagenomic	*Klebsiella*, *Streptococcus*, *Parabacteroides*	*Roseburia*, *Faecalibacterium prausnitzii*		

HT hypertensive; NT, normotensive.

## Data Availability

Not applicable.

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
