# Peer review of "Targeting Gut Microbiota to Treat Hypertension: A Systematic Review"

_ijerph, 2021, doi:10.3390/ijerph18031248_

Round 1

Reviewer 1 Report

The main aim of the present review manuscript was to summarize updated available data on the relation between the gut microbiota changes and blood pressure. This review manuscript is well written and contributes to update knowledge on the gut microbiota variation and its potential effects on blood pressure.

Minor comments

In lines 125-127, the authors stated "Consistently, long-term administration of L. coryniformis and L. gasseri have been demonstrated to reduce systolic BP and reduce vascular inflammation in rats20". The authors should correct this statement as following: "Consistently, long-term administration of Lactobacillus fermentum or L. coryniformis plus L. gasseri have been demonstrated to reduce systolic BP and decrease vascular inflammation in spontaneously hypertensive rats".

Line 133,  the "regradation" word should be corrected.

In lines 255-257, the authors stated "In particular, L. paracasei was negatively, and L. salivarius positively, associated with both BP and dietary sodium (N=829 for 24-hour urinary sodium subsample)5". The authors should check the accuracy of the positive association between L. salivarius and BP.

Author Response

  1. In lines 125-127, the authors stated "Consistently, long-term administration of coryniformis and L. gasseri have been demonstrated to reduce systolic BP and reduce vascular inflammation in rats20". The authors should correct this statement as following: "Consistently, long-term administration of Lactobacillus fermentum or L. coryniformis plus L. gasseri have been demonstrated to reduce systolic BP and decrease vascular inflammation in spontaneously hypertensive rats".

    Reply: We have corrected the statement as the reviewer suggested.

  2. Line 133,  the "regradation" word should be corrected.

    Reply: We have corrected the word to “degradation”.

  3. In lines 255-257, the authors stated "In particular, paracasei was negatively, and L. salivarius positively, associated with both BP and dietary sodium (N=829 for 24-hour urinary sodium subsample)5". The authors should check the accuracy of the positive association between L. salivarius and BP.

    Reply: The association between L. salivarius and BP was studied in Palmu et al. (2020). The abundance of observed bacterial plasmid of L. salivarius was positively associated with pulse pressure. A negative association was observed for L. paracasei with systolic BP, diastolic BP and mean arterial pressure. We have changed the text to indicate that the association between L. salivarius and BP is observed only for pulse pressure.

    Results (line 286-289): “In particular, L. paracasei was negatively associated with BP, whereas L. salivarius was positively associated with only pulse pressure. In addition, L. Paracasei was negatively, and L. salivarius was postitively associated with dietary sodium intake (N=829 for 24-hour urinary sodium subsample).”

Reviewer 2 Report

In this study, Palmu et al. conducted a systematic review focusing on the gut microbiota as a potential and possible therapeutics for hypertension treatment. Since treatment of hypertension is crucial for ameliorating the renal and cardiovascular events, the topic is of interest and this review may draw the attention of a number of readers. However, there are few concerns need to be addressed.

  1. The meaning of the sentence in lines 123-124 “Animal models have also been used to expose hypertensive rodents to probiotics” is not clear.
  2. Lines 127-128, “Increased nitric oxide production may play a critical role in the antihypertensive effect of Lactobacilli”. Rather than increased, reduced nitric oxide production plays a role in antihypertensive effects of Lactobacilli.
  3. Lines 131-133, “In rats, microbial pathways characterized with biosynthesis were upregulated at active phase of day while metabolite regradation pathways were upregulated at rest”. Here ‘regradation’ should be ‘degradation’. Since rats are nocturnal animals, the active phase of rat is considered as ‘night time’ while the resting phase is ‘day time’.
  4. Lines 178-180, “To assess the role of SCFAs in host, careful consideration is necessary to differentiate the associations reported for SCFA producing bacteria, fecal SCFA levels, and circulating (bioactive) SCFA levels”. This statement is better to mention in the “future directions”.
  5. Lines 181-182, “Angiotensin II type 1 receptor treatment of SHRs has been reported to increase fecal acetate, propionate, and butyrate levels”. Here, ‘Angiotensin II type 1 receptor treatment’ should be ‘Angiotensin II type 1 receptor blocker’.
  6. Lines 184-187, “Acetate and butyrate treatment has been reported to alleviate the gut-wall pathology and lower BP in hypertensive rats and mice while no changes were observed in normotensive rats.26,32” Here reference no. 32 is a clinical paper, therefore the citation should be corrected with an appropriate one.   
  7. Lines 203-204, the changes in gut microbiota in hypertension have been associated with increased TH17/Treg ratio in mesenteric lymph nodes that could serve as a marker for increased vascular oxidative stress and endothelial dysfunction.” How come the ratio of TH17/Treg in mesenteric lymphnode serve as a marker for oxidative stress in vasculatures? This reviewer suggests to go through the cited paper and explain the information appropriately from that original paper.
  8. Lines 265-266, “SCFAs are bacterial metabolites that are detected in host circulation by G protein- coupled receptors”. Since the authors explained the SCFAs earlier, rather than giving definition of SCFAs here, better to give the definition in the earlier part of the manuscript where it has mentioned for the first time.
  9. Lines 291-292, “one interventional pilot human performed in humans.” The meaning of this sentence is not clear.
  10. General comments for Table 1 and Table 2
    - All the cited reference numbers are incorrect with those of information stated in the tables. May be something wrong in the reference management system. Attention should be required.
    - Regarding the animals, please use the similar name for a specific animal model consistently. Dahl rats, Dahl salt-sensitive rats, salt-sensitive Dahl rats all are similar. It is better to mention first “Dahl salt-sensitive (DSS) rats” and then “DSS” in the subsequent rows. Similar for germ free and GF mice.
    -Regarding intervention, “gavage treatment” is not an intervention rather it is a method of drug administration orally. Therefore, in the place of gavage treatment (Adnan et al, Li et al, Mell et al, Toral et al.) please provide the real name of the drugs or intervention.
  11. In Table 1, Adnan et al. study conducted on WKY and SHR, where SHR was the hypertensive animals (study subjects) and WKY was the normotensive control. So, rather that WKY, mentioning SHR would be appropriate.
  12. In Table 1, the intervention for Chakraborty et al. study would be “diurnal timing dependent alterations or disrupted circadian rhythm”.
  13. In Table 1, the “decreased or downregulated in serum, plasma or vasculatures” column for G’omez-Guzm’an et al and Marques et al studies contain the information regarding heart weight, heart rate, cardiac fibrosis and left ventricular hypertrophy. This reviewer suggests to arrange these information in an appropriate way. 
  14. Care should be taken for typo or grammatical errors.  

Author Response

  1. The meaning of the sentence in lines 123-124 “Animal models have also been used to expose hypertensive rodents to probiotics” is not clear.

    Reply: We have made the sentence more comprehensible.

    Results (124-126): “The association between probiotic intake and blood pressure has been studied in experimental animal models.”

  2. Lines 127-128, “Increased nitric oxide production may play a critical role in the antihypertensive effect of Lactobacilli”. Rather than increased, reduced nitric oxide production plays a role in antihypertensive effects of Lactobacilli.

    Reply: Robles-Vera (2018) reported that “a chronic LC40 treatment, at a dose that reduced BP in SHR, was unable to prevent the raise in BP and the hypertrophic effects in heart and kidney induced by blockade of NO synthesis, despite the improvement of vascular oxidative stress and inflammation.” Therefore, the antihypertensive role of LC40 appears to be connected with increased NO synthesis. We have formulated the sentence to make the sentence easier to understand (increasing beneficial effect vs. decreasing harmful effect).

    Results (129-130): “Increased nitric oxide production may play a critical role in the beneficial (antihypertensive) effect of Lactobacilli.”

  3. Lines 131-133, “In rats, microbial pathways characterized with biosynthesis were upregulated at active phase of day while metabolite regradation pathways were upregulated at rest”. Here ‘regradation’ should be ‘degradation’. Since rats are nocturnal animals, the active phase of rat is considered as ‘night time’ while the resting phase is ‘day time’.

    Reply: We have corrected the word to degradation. We also now state more clearly that night time is the active time for rats to avoid confusion.

    Results (133-135): “In rats, microbial pathways characterized with biosynthesis were upregulated at active phase of day (nighttime) while metabolite degradation pathways were upregulated at rest (daytime).”

  4. Lines 178-180, “To assess the role of SCFAs in host, careful consideration is necessary to differentiate the associations reported for SCFA producing bacteria, fecal SCFA levels, and circulating (bioactive) SCFA levels”. This statement is better to mention in the “future directions”.

    Reply: We have moved the statement to the future directions section as the reviewer suggested.

  5. Lines 181-182, “Angiotensin II type 1 receptor treatment of SHRs has been reported to increase fecal acetate, propionate, and butyrate levels”. Here, ‘Angiotensin II type 1 receptor treatment’ should be ‘Angiotensin II type 1 receptor blocker’.

    Reply: We have now corrected the name of the medication group.

    Results (lines 198-199): “Angiotensin II type 1 receptor blocker treatment of SHRs has been reported to increase fecal acetate, propionate, and butyrate levels.”

  6. Lines 184-187, “Acetate and butyrate treatment has been reported to alleviate the gut-wall pathology and lower BP in hypertensive rats and mice while no changes were observed in normotensive rats.26,32” Here reference no. 32 is a clinical paper, therefore the citation should be corrected with an appropriate one.

    Reply: Kim et al. (2018) also studied in their manuscript whether butyrate supplementation would ameliorate high BP in mice chronically infused with Angiotensin II. They reported that Butyrate co-administration to C57Bl6 mice infused with Angiotensin II significantly reduced MAP and prevented the alterations in the levels of mRNA for occludin, ZO-1, and claudin 4 by qPCR in Ang II mice intestine. Therefore, reference 32 also includes results from animal models.

  7. Lines 203-204, the changes in gut microbiota in hypertension have been associated with increased TH17/Treg ratio in mesenteric lymph nodes that could serve as a marker for increased vascular oxidative stress and endothelial dysfunction.” How come the ratio of TH17/Treg in mesenteric lymphnode serve as a marker for oxidative stress in vasculatures? This reviewer suggests to go through the cited paper and explain the information appropriately from that original paper.

    Reply: We have improved the two paragraphs that review the work by Toral et al. (2019) to make the thought process more logical.

    Results (lines 228-234): “The changes in gut microbiota in hypertension have been associated with increased TH17/Treg ratio in mesenteric lymph nodes and activation of adaptive immune response.14 The B7 ligand costimulated T cell activation and modulation of TH17/IL17 axis have been proposed to share an essential role in the development of endothelial dysfunction, increased vascular oxidative stress, and hypertension upon fecal microbiota transplantation from SHR to Wistar Kyoto rats.14

  8. Lines 265-266, “SCFAs are bacterial metabolites that are detected in host circulation by G protein- coupled receptors”. Since the authors explained the SCFAs earlier, rather than giving definition of SCFAs here, better to give the definition in the earlier part of the manuscript where it has mentioned for the first time.

  9. Reply: We now explain SCFAs when their role is reviewed first time in the animal model section. We agree that repetition should be avoided and have removed the indicated sentence.

  10. Lines 291-292, “one interventional pilot human performed in humans.” The meaning of this sentence is not clear.

    Reply: We have clarified the unclear sentence.

    Results (331-333): ”These results are in part supported by results from four large-scale observational epidemiological studies and from one interventional pilot study performed in humans.”

  11. General comments for Table 1 and Table 2
    - All the cited reference numbers are incorrect with those of information stated in the tables. May be something wrong in the reference management system. Attention should be required.
    - Regarding the animals, please use the similar name for a specific animal model consistently. Dahl rats, Dahl salt-sensitive rats, salt-sensitive Dahl rats all are similar. It is better to mention first “Dahl salt-sensitive (DSS) rats” and then “DSS” in the subsequent rows. Similar for germ free and GF mice.
    -Regarding intervention, “gavage treatment” is not an intervention rather it is a method of drug administration orally. Therefore, in the place of gavage treatment (Adnan et al, Li et al, Mell et al, Toral et al.) please provide the real name of the drugs or intervention.

    Reply: We thank the reviewer for the sharp observation and have now corrected the citation numbers in the tables. We also agree that consistent terminology would improve the manuscript. We agree also that correct intervention name should be used in the table instead of the “gavage treatment”. All of these issues have now been corrected in Tables 1 and 2.

  12. In Table 1, Adnan et al. study conducted on WKY and SHR, where SHR was the hypertensive animals (study subjects) and WKY was the normotensive control. So, rather that WKY, mentioning SHR would be appropriate.

    Reply: Adnan et al. (2016) used cecal content from stroke prone rats (SHRSP) and normotensive WKY rats. Then four-week-old WKY and SHR rat were gavaged with WKY or SHRSP microbiota, resulting in four groups. For summary table we considered to select either WKY with SHRSP microbiota (WKY g-SHRSP) or SHR with WKY microbiota (SHR g-WKY). We thought that reporting the results for initially normotensive rats would be the more interesting one highlighting pathogenesis instead of therapeutics.

  13. In Table 1, the intervention for Chakraborty et al. study would be “diurnal timing dependent alterations or disrupted circadian rhythm”.

    Reply: We have changed the column description as the reviewer proposed.

  14. In Table 1, the “decreased or downregulated in serum, plasma or vasculatures” column for G’omez-Guzm’an et al and Marques et al studies contain the information regarding heart weight, heart rate, cardiac fibrosis and left ventricular hypertrophy. This reviewer suggests to arrange these information in an appropriate way. 

    Reply: We aimed to balance the readability, provided information, and dimensional limitations in the two large tables in the review; ultimately, we were unable to fit separate column for these features. We have now changed the column title to better reflect the contents.

    Table 1 column name: “Increased or uppregulated in serum, plasma, vasculature, or organs” and “Decreased or downregulated in serum, plasma,  vasculature, or organs”

  15. Care should be taken for typo or grammatical errors.  

    Reply: We are sorry for the typos and grammatical errors. We have now gone through the text again and corrected all typos and grammatical errors that we could observe.

Reviewer 3 Report

In this manuscript, Palmu et al. provide a systematic review of relationships between gut microbiota and hypertension in animals and humans. This is a valuable review of the above topic.  This review is well conceived and structured, and provides current knowledge in this area. This reviewer has only a few minor comments listed below.

  1. P4, para4, line181. Please mention that the numbers of acetate- and butyrate-producing bacteria were reported to be decreased in SHRs before “Angioensin II type…”.
  2. P4, para5. Please elaborate on the seminal work of Pluznick et al. (ref 29), who revealed that SCFA produced by the gut microbiota modulate blood pressure via host G-protein coupled receptors, including OLfr78 and Gpr41.
  3. It is also advisable to discuss the relationship between the Firmicutes/Bacteroidetes rate and blood pressure.
  4. P5, para2, line198 & P5, para 3, line212. Please clarify the concentration of salt used and the duration of the treatment period.
  5. P6, para2, line250. Again, please clarify the duration of the treatment period. For two weeks?
  6. Are there any literature that tested the longer term effect of high-salt diet on gut microbiota composition and blood pressure?

Author Response

  1. P4, para4, line181. Please mention that the numbers of acetate- and butyrate-producing bacteria were reported to be decreased in SHRs before “Angioensin II type…”.

    Reply: We agree that information about SCFA producing bacteria would improve the paragraph and have included the proposed mention.

    Results (lines 197-198): “Hypertension has been associated with a decreased number of acetate- and butyrate-producing bacteria.21

  2. P4, para5. Please elaborate on the seminal work of Pluznick et al. (ref 29), who revealed that SCFA produced by the gut microbiota modulate blood pressure via host G-protein coupled receptors, including OLfr78 and Gpr41.

    Reply: We agree that the seminal work of Pluznick et al. should be introduced better and this information would make the manuscript more easily readable.

    Results (lines 205-206): “SCFAs have been demonstrated to affect renin release in juxtaglomerular cells and modulate BP through G-protein coupled receptors.29

  3. It is also advisable to discuss the relationship between the Firmicutes/Bacteroidetes rate and blood pressure.

    Reply: The Firmicutes/Bacteroidetes ratio has been reported as a marker of gut dysbiosis even in recent articles. However, due to the nature of ratios, the change of Firmicutes/Bacteroidetes ratio does not contain information about which one of the two taxa changed. Firmicutes and Bacteroidetes are also high-level taxa in phylogenetic tree (Phylum) and therefore offers a heterogenic and broad measure (coarse) of the gut microbiota. We have now discussed the challenges of assessing the association between Firmicutes/Bacteroidetes ratio and blood pressure in the manuscript but do not focus on this domain in more detail.

    Results (lines 334-338): “Although several studies have reported on the potential association between the Firmicutes/Bacteroidetes ratio and blood pressure,13,19,22,26,28,31,37,43 this approach has several challenges. These challenges include the lack of information on which of the two taxa is the main driver of the ratio and the use of relatively heterogenic and coarse phylum-level information. We have therefore refrained from focusing on these studies in this review.”

  4. P5, para2, line198 & P5, para 3, line212. Please clarify the concentration of salt used and the duration of the treatment period.

    Reply: We have included the information about length and concentration of dietary salt.

    Results (lines 222-226): “High salt intake has been reported to modulate gut microbiota particularly depleting L. murinus in mice (4 % dietary and 1 % drinking water NaCl vs. 0.5 % dietary NaCl for 14 days) while the growth of various Lactobacilli was inhibited by sodium (half maximal growth inhibition 0.6 mol/l in vitro vs. 0.3 mol/l colonic NaCl concentration under high dietary salt).17

    Results (lines 236-238): “In addition to the decrease of Lactobacillus species, various gut microbial changes have been associated with high dietary salt (4-8 % dietary NaCl; Table 1).18,35–37

  5. P6, para2, line250. Again, please clarify the duration of the treatment period. For two weeks?

    Reply: Two weeks is correct; we have included the information about the duration of the intervention in the paragraph.

    Results (lines 279-282): “To study the effect of high-salt diet on gut microbiota, a pilot study (N=12) was performed in healthy males receiving slow-releasing 6 g sodium chloride supplementation (total salt intake 13.8±2.6 g/day for 14 days).17

  6. Are there any literature that tested the longer term effect of high-salt diet on gut microbiota composition and blood pressure?

    Reply: To our best knowledge we have reviewed the available literature and longer term effect of high-salt diet is not yet available. In humans, as the accurate long-time measurement of dietary salt is challenging (multiple 24 h urine collections are needed for measurements [10.3945/ajcn.116.132951]) and repeated dietary questionnaires are difficult to perform in an uncontrolled setting. Animal models might provide us more information about this in the future.